# Preparation and Electrocatalytic Activity of a Cobalt Mixed Nitrogen 3D Carbon Nanostructure @ Carbon Felt toward an All-Vanadium Redox Flow Battery

**Jun Su \*, Zongyang Li, Longlong Hao \* and Lilu Qin**

School of Materials Science and Engineering, Chongqing University, Chongqing 400044, China; lizongyang@cqu.edu.cn (Z.L.); qinlilu@cqu.edu.cn (L.Q.)

\* Correspondence: sujun@cqu.edu.cn (J.S.); haolonglong@cqu.edu.cn (L.H.); Tel.: +86-023-65106001 (J.S.); +86-188-833-45383 (L.H.)

**Abstract:** All-vanadium redox flow batteries (VRFBs), with good operation flexibility and scalability, have been regarded as one of the most competitive substitutes for large-scale energy storage. However, because of the low electrochemical activities of traditional electrodes such as carbon felt and graphite felt, they will impede the interfacial charge transfer processes and decrease the efficiencies of VRFBs. In this work, Co-MOF (ZIF-67) was prepared as a precursor, and a cobalt mixed nitrogen 3D carbon nanostructure and carbon felt (Co-CN@CF) was prepared by chemical reaction and used in VRFBs as electrodes. With the unique structure and high efficiency catalyst on the carbon felt, the Co-CN@CF exhibited excellent electrochemical activity toward the $VO^{2+}/VO_2^+$ redox couple in the VRFB, with an average cell voltage efficiency (VE) of 86% and an energy efficiency (EE) of 82% at 80 mA cm$^{-2}$, which was increased by more than 10% compared with the traditional carbon felt. VRFBs with a Co-CN@CF electrode also showed much better long-term stability (over 1000 cycles) compared with the battery with a pristine CF electrode.

**Keywords:** MOF; redox flow batteries; electrode; carbon felt; nitrogen 3D carbon nanostructure; electrocatalytic activity; cobalt

## 1. Introduction

With the increasing awareness of energy conservation and environmental protection, it is important to use renewable energy sources such as solar, wind, and hydropower to replace fossil fuels. However, renewable energy is always unstable and inconsecutive, and the intermittence may open spatial and temporal gaps between the availability of the energy and its consumption by end users [1]. In order to address these issues, it is necessary to develop suitable energy storage systems for the power grid [2–5]. There are many electrochemical energy storage (EES) systems, such as lithium-ion, sodium–sulfur, metal–air, and all-vanadium redox flow batteries (VRFBs). In particular, in comparison with other EES, the VRFB system can prove a few hundred kW and even multi-mW levels for load leveling or peak shaving [6]. They have been widely used due to their advanced properties, including fast response, long life, flexible operation, and environmental friendliness [7–10]. The types of redox flow batteries are determined by the redox active species that occur at electrode, such as Zn/Br$_2$, Fe/Cr, Zn/polyiodide, polysulfide/Br$_2$, and so on [11–15]. Among these batteries, the VRFB, which was invented in the 1980s, is a promising candidate, with its high reliability, deep-discharge capability, high energy efficiency, and electrochemical reversibility due to employing the same vanadium metal in both the positive and negative half-cells in different oxidation states.

Vanadium redox reactions have been considered as a key factor affecting the energy efficiency of the all-vanadium redox flow batteries (VRFBs). VRFBs based on V(IV)/V(V) and V(II)/V(III) redox couples can provide an open circuit voltage of 1.26 V, respectively, as follows [16–18]:

Positive electrode:

$$VO_2^+ + 2H^+ + e^- = VO^{2+} + H_2O; \quad E^0 = 1.0 \text{ V vs. SHE} \tag{1}$$

Negative electrode:

$$V^{2+} = V^{3+} + e^-; \quad E^0 = -0.26 \text{ V vs. SHE} \tag{2}$$

Overall reaction:

$$VO_2^+ + V^{2+} + 2H^+ = VO^{2+} + V^{3+} + H_2O; \quad E = 1.26 \text{ V} \tag{3}$$

As shown in the center of Figure 1, a VRFB consists of the electrodes, a membrane, and the electrolyte. The electrodes, with high electrical conductivity and stability in concentric acid-based vanadium electrolytes, can control the electron transfer and mass transport [19]. Materials such as graphite felt (GF), carbon felt (CF), and carbon paper (CP) are promising electrodes for VRFBs, as they are characterized by a low cost and wide range of operating potential. However, those materials with low electrochemical activity, poor kinetic reversibility and wettability, and a small amount of surface defects and functional groups are not enough for practical application. To solve these problems, many researchers have studied many surface modifications of carbon felt to improve its electrochemical performance and catalytic activity. One of the surface modifications is direct surface treatment to introduce functional groups on CF or GF, including acid treatment [20], thermal treatment [21], KOH activation [22], and electrochemical oxidation [23]. These treatments could influence the polarization effects in three ways: concentration polarization, ohmic polarization, and electrochemical polarization [24]. In general, before the treatment mentioned above, the carbon electrodes are hydrophobic, and after treatment, they become hydrophilic. This is beneficial for aqueous-based RFBs. Coating and depositing other metal or carbon derivatives onto CF or GF can also improve their properties. The metals (Pt, Ir, Bi, Ru, Cu, etc.) could improve the electron transfer rate in VRFBs [25–28]. In the early 1990s, electrodeposition and thermal reduction of various precious metals on carbon materials were introduced to increase the electrical conductivity. In addition, the metalized CFs by ion exchange of $PT^{4+}$, $Au^{4+}$, $Mn^{2+}$, $In^{3+}$, and $Ir^{3+}$ in solution were studied to prove their behavior in VRFBs. However, the high cost limited the development of metalized CF electrodes. Therefore, some low-cost transition metal–oxide electrocatalysts such as $Mn_3O_4$, $PbO_2$, $WO_3$, $Nb_2O_5$, $CeO_2$, and $IrO_2$ have been substituted for noble metals to improve the catalytic activity of VRFBs. However, this kind of electrode is unstable in an acid electrolyte [29–31].

Recently, carbon-based electrodes such as activated carbon (AC), carbon nanoparticles, carbon nanotubes, and graphene-based nanosheets have been widely studied. These can improve the specific surface area of a CF electrode and catalyze the redox reaction, but its resistance will be higher. So, the combination of carbon-based materials with metal elements has been studied in order to effectively catalyze the vanadium redox couples. For example, platinum with high electric conductivity and stability was mixed with carbon black and graphene. Tseng et al. [32–34] investigated the reaction rate of VRFBs by PT/C (Vulcan XC-72). It was reported that the Co-based metal–organic framework (Co-MOF) ZIF-67 was a perfect carbon material for lithium batteries due to its porous structure. GE et al. [35] reported on a GF electrode that was pasted with CoP that could effectively promote the V(IV)/V(V) redox reaction, and the CoP power was synthesized by direct carbonization of ZIF-67.

ZIF-67 is a Co-based metal-organic framework (MOF) consisting of metal ions or clusters coordinated with organic ligands to form one-, two-, or three=dimensional structures. MOFs are a subclass of coordination polymers with porous features. Due to their special structure and metallic characteristics, MOFs are used in gas purification, gas separation, and catalysis, and as sensors and supercapacitors [36,37].

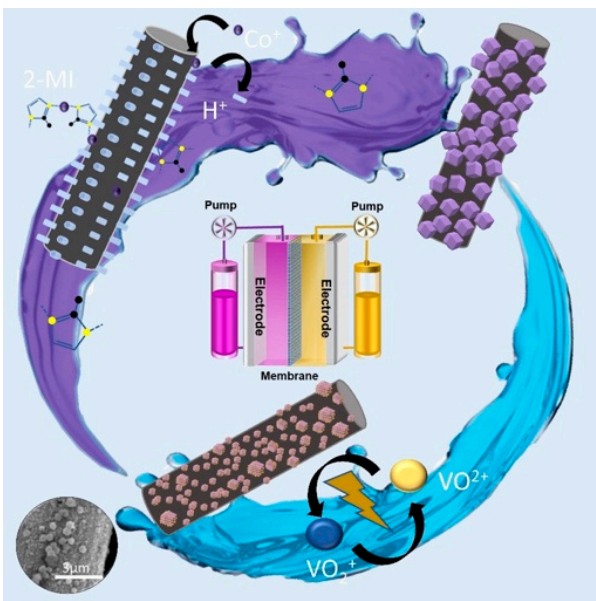

**Figure 1.** The synthesis process of ZIF-67@CF and Co-CN@CF.

In this paper, a novel type of metal-carbon composite on CF was prepared, and it contributed to the performance improvement of VRFBs. Firstly, ZIF-67 was grown in situ on CF as a precursor, in a new attempt to replace the traditional postbonding process. Then, the ZIF-67@CF was treated in a vacuum oven with inert gas, and a cobalt mixed nitrogen 3D carbon nanostructure with carbon felt (Co-CN@CF) was prepared. With this method, the as-prepared Co-CN particles on the CF maintained the morphology of ZIF-67, which is a regular dodecahedron. The Co-CN@CF was used as a novel catalyst in a VRFB to increase the specific surface area and improve the electrochemical catalysis capability of CF. The results showed that the Co-CN@CF exhibited excellent electrochemical activity toward $VO^{2+}/VO_2^+$ redox couples in the VRFB, with an average cell voltage efficiency (VE) of 86% and an energy efficiency (EE) of 82% at 80 mA cm$^{-2}$, which was increased by more than 10% compared with the traditional carbon felt.

## 2. Materials and Methods

### 2.1. Synthesis of ZIF-67@CF and Co-CN@CF

The ZIF-67@CF was firstly synthesized. Typically, the carbon felts (Jingu Carbon Material Co., Ltd. Liaoning, China) were immersed in concentrated sulfuric acid (H$_2$SO$_4$) for 2 h, rinsed with deionized water 3 times, and dried at 80 °C for 20 h. After that, 1.436 g (4.9 mmol) of cobalt nitrate hexahydrate (Co (NO$_3$)$_2$·6H$_2$O) and 3.244 g (39.5 mmol) of 2-methylimidazole (2-MI) were dissolved in 100 mL methanol. The carbon felt (CF) was immersed in the methanol solution with cobalt nitrate hexahydrate under vigorous stirring for 15 min. After that, the two solutions were mixed together quickly under magnetic stirring for 20 min. The solution was kept static for 24 h (5, 12, and 20 h were also used) at room temperature. The product (denoted as ZIF-67@CF) was collected, washed with ethyl alcohol, and dried at 80 °C for 24 h.

Finally, the ZIF-67@CF was heated to 600, 700, 800, and 900 °C for 2 h at a rate of 2 °C/min$^{-1}$ under an Ar atmosphere. The ZIF-67@CF changed into Co-CN@CF.

### 2.2. Material Characterization

Scanning electron microscopy (SEM) and energy-dispersive spectrometer (EDS) micrographs were collected on a FEI NOVA 400. X-ray diffraction analysis (XRD, D/max 2500) using Cu-Ka radiation with an incidence beam angle of 2° and a 2θ degree range of 10–90° was used to investigate the crystalline phase in the samples. X-ray photoelectron

spectroscopy (XPS, ESCA-LAB250Xi) was used to explore the surface chemistry of the as-developed samples.

### 2.3. Electrochemical Characterization

The Co-CN@CF was cut at a thickness of 0.5 mm and an area of $1 \times 2$ cm$^2$. Cyclic voltammetry (CV) and electrochemical impedance spectroscopy (EIS) tests were used to evaluate the electrochemical characterization of the Co-CN@CF electrodes on an electrochemical workstation (Chenhua 406E). The CV and EIS test were carried out in a three-electrode configuration with a $2 \times 2$ cm$^2$ Pt and a mercurous sulfate electrode with the electrolyte solution (0.1 mol/L VOSO$_4$ in 3 mol/L H$_2$SO$_4$). For the CV test, the scanning range varied from 0 to 0.8 V at a rate of 5 mV/s. The EIS test was measured under open circuit potential (OCP) with a potential amplitude of 5 mV at a frequency ranging from 1 M to 0.1 Hz.

### 2.4. Single-Cell Test

The electrochemical performance of the electrodes was evaluated by constant-current charge–discharge experiments using a VRFB on a CT5000A (LAND, Wuhan, China) battery test system. The cell used 15 mL of 1.5 mol/L V(III)/V(IV) in 3 mol/L H$_2$SO$_4$ as the electrolyte, Nafion 212 (DuPont, America) as the membrane, CF ($2 \times 2$ cm$^2$) as the negative electrode, and a CF ($2 \times 2$ cm$^2$) or Co−CN@CF ($2 \times 2$ cm$^2$) as the positive electrode. The flow rate of the pump was 40 mL/min at room temperature. The scan potential range of the test was 0.6 to 1.75 V, and the rate capacity of the test was examined at a current density of 30–100 mA cm$^{-2}$.

## 3. Results and Discussion

The precursor ZIF-67 was grown in situ on CF, which was different from the traditional postbonding method. In order to study the crystallization course of the ZIF-67, the SEM images of the ZIF-67 at different reaction times (0, 5, 12, and 20 h) were collected, as shown in Figure 1a–d. Firstly, Co$^{2+}$ ions replaced the H$^+$, which combined with the CF and captured the 2-methylimidazole surrounding it in the solution, and many buds grew in situ on the surface of the CF (Figure 1b). This was the growing point for the ZIF-67, and determined the in situ growth state and how the ZIF-67 bonded with the CF. With the reaction proceeding, an increasing amount of 2-methylimidazole was absorbed on the surface, and bridged as shown in Figure 1c. Finally, the Co$^{2+}$ ions bonded with the 2-methylimidazole to form 3D structures, such as diamond 12-face bodies and cubes. Those structures were called ZIF-67, and the as-treated CF was called ZIF-67@CF.

In order to determine the changes to the CF surface, the SEM images of pristine CF, ZIF-67@CF, and Co-CN@CF were measured; these are shown in Figure 2. In the SEM images of ZIF-67, it can be seen that the original Co-MOF precursor had a regular shape, and its particle size was about 200–300 nm. In addition, the EDS of the ZIF-67 confirmed that the Co-MOF was evenly distributed on the surface. After the ZIF-67@CF was heated to 600 °C for 2 h at a rate of 2 °C·min$^{-1}$ under an Ar atmosphere, the SEM image showed that some of the product particles nearly maintained the morphology of the original MOF, and some product particles became smaller, in the manner of a glass splinter adhering to the CF. This proved that carbonization of ZIF-67 at a high temperature led to the decrease in the particles' size.

The XRD was used to analyze the phase composition of the ZIF-67, ZIF-67@CF, and Co-CN@CF. The XRD of the ZIF-67 powder centrifugal separated from the solution showed the ZIF-67@CF reaction (Figure 3a). The results also indicated that the as-prepared Co-MOF agreed with the simulated results, demonstrating that the as-prepared Co-MOF had a crystalline feature. The XRD of the ZIF-67@CF and Co-CN@CF (Figure 3b,c) showed a strong peak at around 25° for both the ZIF-67@CF and Co-CN@CF that was assigned to the (002) reflection of carbon. The other peaks were in agreement with ZIF-67 for the Co-CN@CF, while the other peaks were in agreement with the Co standard pattern (PDF

Card No. 15-0806). This meant that during the heating process, organic ligands and some absorbed compounds in the ZIF-67 were decomposed to form a C mixed N structure, while the residual Co$^{2+}$ ions were changed to Co metallics. No impurity peaks appeared, implying that the product was pure Co metallics.

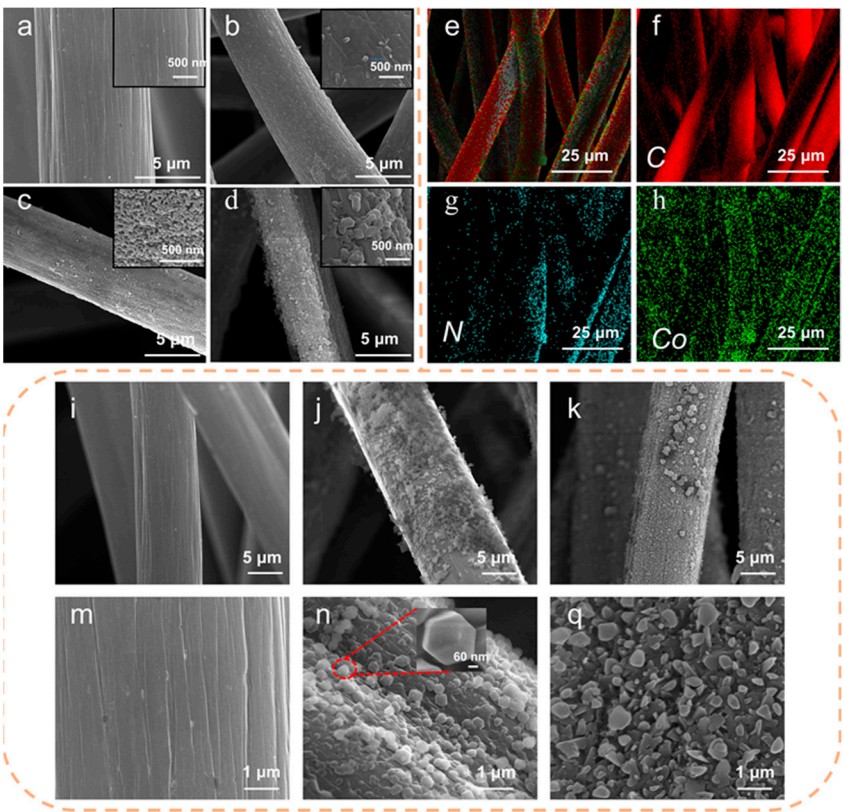

**Figure 2.** SEM images of ZIF-67 at different times: (**a**) before treatment; (**b**) after 5 h; (**c**) after 12 h; (**d**) after 20 h. (**e**–**h**) EDS images of the ZIF-67@CF. SEM images of: (**i**,**m**) CF; (**j**,**n**) ZIF-67@CF (after 24 h); (**k**,**q**) Co-CN@CF.

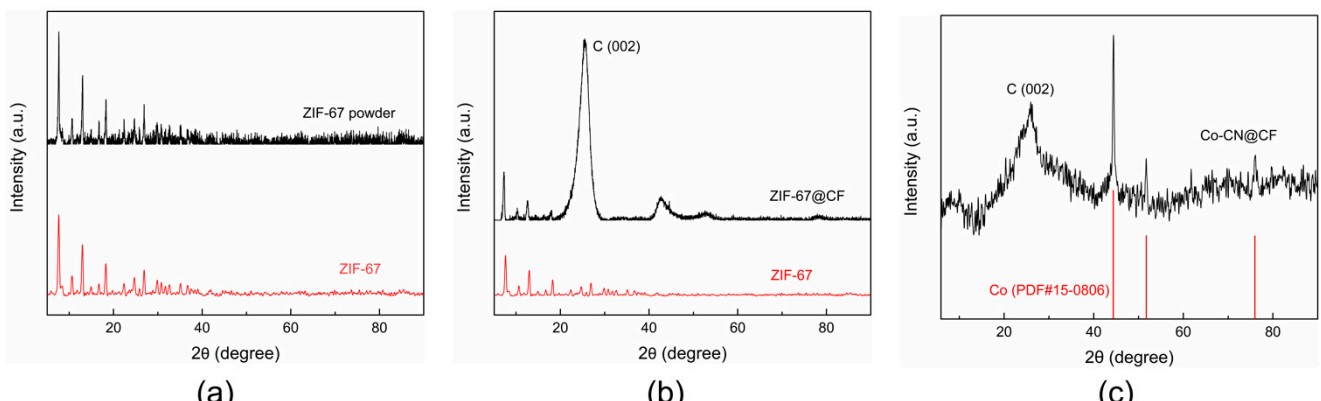

**Figure 3.** XRD results: (**a**) ZIF-67 powder; (**b**) ZIF-67@CF; (**c**) Co-CN@CF.

The XPS curves are shown in Figure 4. We determined that the functional group of the ZIF-67@CF was changed, and the Co metal element was combined with the CF. As the surveys (Figure 4a,f) of all chemical elements showed, the peaks of the Co-CN@CF were different from those of the ZIF-67@CF; there were Co 3s and Co 3p peaks in the curves of the Co-CN@CF, and the peak intensities for Co and O in the curves of Co-CN@CF were higher than those in the curves of ZIF-67@CF. However, the peak intensities for N and C in

the curves of Co-CN@CF were lower than those in the curves of ZIF-67@CF. This meant that the structure of the ZIF-67 changed after annealing, as Co and O were brought onto the surface of CF. As shown in the C 1s (Figure 4b,g) and N 1s (Figure 4a,c,h) peaks, there were only Co-coordinated-N and NSP2-C bonds in the ZIF-67@CF, because the 2-MI reacted with $Co^{2+}$ and formed an MOF structure. After annealing, the Co-coordinated-N and NSP2-C bonds broke down, and C combined with CF, O, and N to form C=C, C−O, and C-N bonds, while only a few Co-coordinated-N bonds were kept. For the Co 2p (Figure 4e,i) peaks and O 1s (Figure 4f,j) peaks, there were Co metals on the surface of the CF that originated from the ZIF-67 MOFs, as the -OH group in the Co-CN@CF could provide more reaction points for electrode reactions.

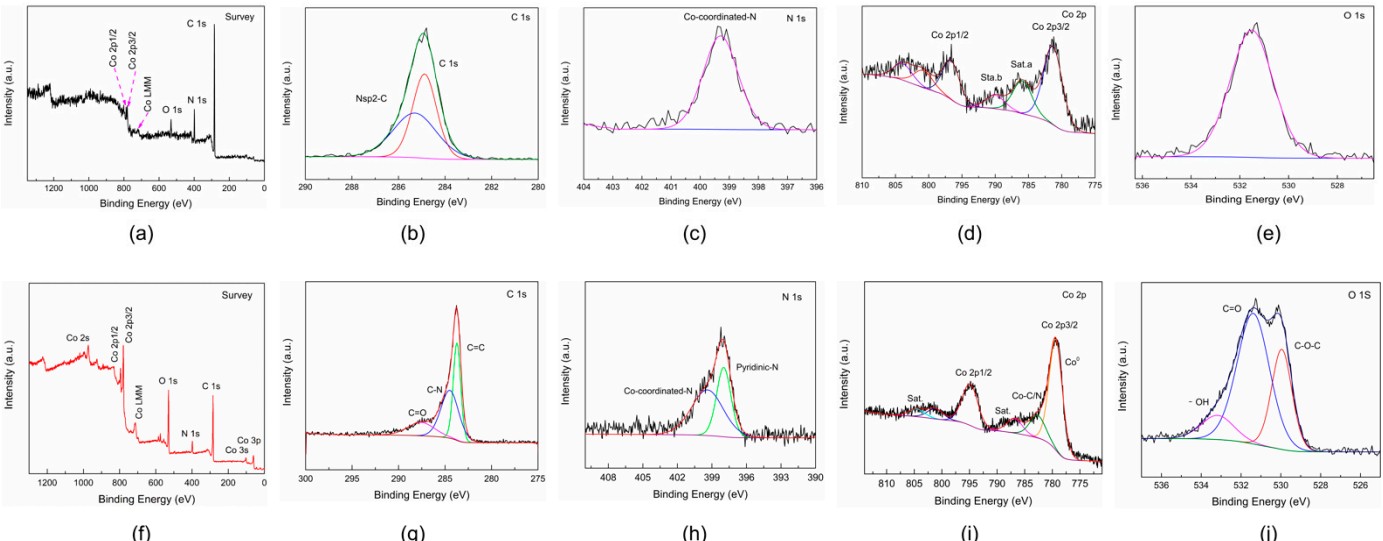

**Figure 4.** XPS results: (**a**) the surveys of all chemical elements in ZIF−67@CF; (**b–e**) the peaks of C 1s, N 1s, Co 2p, and O 1s in ZIF−67@CF, respectively; (**f**) the surveys of all chemical elements in Co−CN@CF; (**g–j**) the peaks of C 1s, N 1s, Co 2p, and O 1s in Co−CN@CF, respectively.

Figure 5a provides CV curves of the CF electrode using different treatment methods. There were two main peaks in all the curves, representing the oxidation and reduction peaks corresponding to the $VO^{2+}/VO_2^+$ couple. The current difference in the peak potential separation between the two oxidation and reduction peaks could be used to evaluate the redox reaction kinetics and reversibility for the $VO^{2+}/VO_2^+$ couple. For the pristine CF electrode, the oxidation and reduction peak potentials appeared at 0.596 and 0.201 V (vs. SCE); the peak potentials of the CF/600 °C electrode appeared at 0.585 and 0.201 V (vs. SCE); the peak potentials of the ZIF-67@CF electrode appeared at 0.58 and 0.194 V (vs. SCE); while the peak potentials of the Co-CN@CF/600 °C electrode appeared at 0.551 and 0.215 V (vs. SCE). It was clear that the peak potential separation of the Co-CN@CF/600 °C electrode (0.33 V) was the smallest among all the electrodes, which indicated a better reversibility of the redox reaction on the Co-CN@CF/600 °C electrode. Moreover, the oxidation and reduction peak current densities of the Co-CN@CF/600 °C electrode were 83.84 and 85.36 mA cm$^{-2}$, which were the highest among those GF electrodes. Moreover, for the $|I_p^a/I_p^c|$ of those electrodes (as shown in Figure 5b), the $|I_p^a/I_p^c|$ of the Co-CN@CF/600 °C electrode and the ZIF-67@CF electrode were higher than 90% and better than the others; the highest one was shown for the Co-CN@CF/600°C electrode (98%), which was close to 100%. The remarkable increases in the peak current density and the $|I_p^a/I_p^c|$ of the Co-CN@CF/600 °C meant that the electrocatalytic activity and reversibility of the CF were enhanced by introducing the ZIF-67 group and Co-CN structure. However, it seriously degraded with a heat treatment temperature higher than 600 °C, such as 900 °C. All the results showed that the ZIF-67 and Co-CN grown in situ on the CF were excellent catalysts for the $VO^{2+}/VO_2^+$ couple.

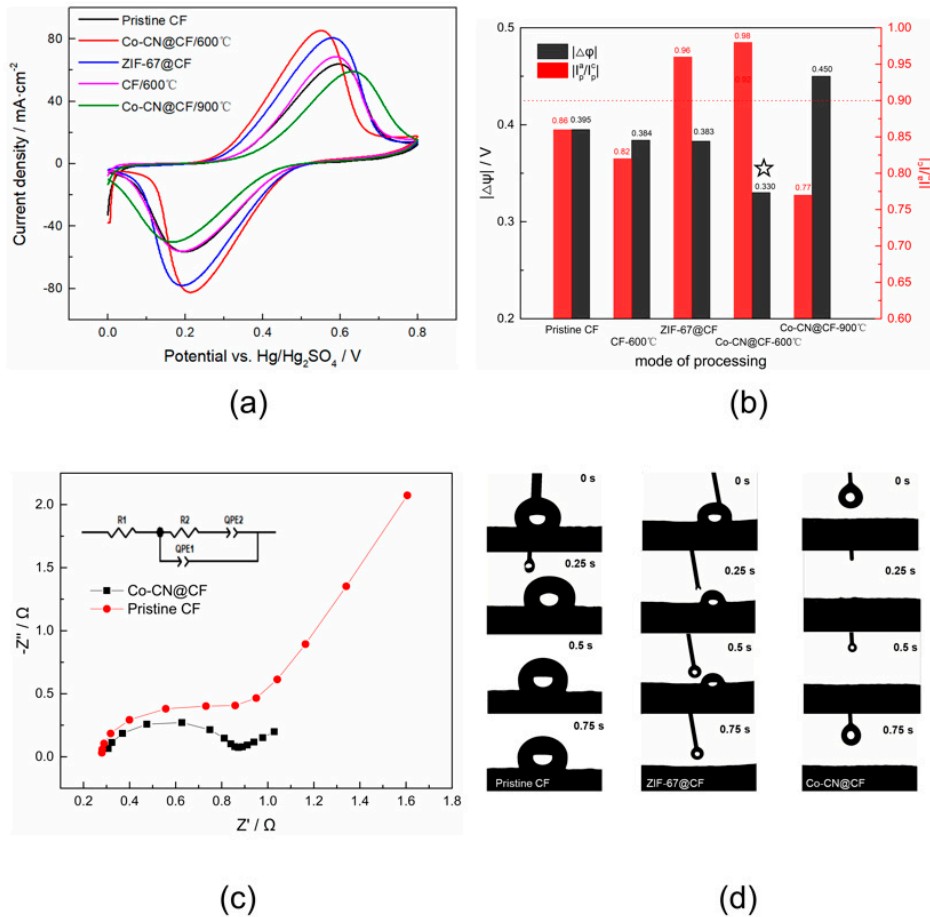

**Figure 5.** (**a**) CV curves of CF electrode using different treatment methods; (**b**) performance analysis of CF electrode using different treatment methods; (**c**) EIS measurement of electrode; (**d**) wettability of different CFs.

EIS measurements were also used to study the electric charge transfer properties of the electrodes. Nyquist plots for the pristine CF and Co-CN@CF electrodes measured at open circuit potential (OCP), as well as in the corresponding equivalent circuit based on measured data, are shown in Figure 5c. The Nyquist plots for the two electrodes maintained the same shape, including a high-frequency semicircle and a low-frequency sloping line. For the equivalent circuit, there were two resistances, R1 and R2. R1 was the sum of the electrolyte solution and electrode resistance, while R2 represented the charge transfer resistance in the interface between the electrolyte solution and electrode. Q1 and Q2 were the constant-phase elements, representing the diffusion capacitance caused by $VO^{2+}$ and $VO_2^+$ diffusion and electric double-layer capacitance in the electrode–solution interface, respectively. According to the fitting results, the R1 for the pristine CF and the Co-CN@CF were almost the same (0.29 $\Omega$), which meant that the two electrodes had a similar resistance. However, the R2 value of the Co-CN@CF (0.56 $\Omega$) was lower than that of the pristine CF (0.68 $\Omega$), which meant that the charge transfer resistance of the Co-CN@CF was lower, and the structure could accelerate the electron transfer.

The wettability of the Co-CN@CF and ZIF-67@CF was better than that of the pristine CF. As shown in Figure 5d, the contact angle of the pristine CF was >90°, and the Co-CF@CF and ZIF-67@CF were superhydrophilic. With the increase in the -OH functional group on the surface of CF, the wettability of the CF increased, and the wettability of the Co-CN@CF was better than that of the ZIF-67@CF. This caused the electrolyte to react on the CF faster and more efficiently.

A charge–discharge test and rate performance of the single cells were implemented in a redox flow battery (Figure 6d) that was designed by our laboratory in order to further

understand the effect of the Co-CN catalyst on the electrochemical performance of the CF. As shown in Figure 5a, lifetime tests were further carried out to study the long-term stability of the cells using different membranes. The tests were terminated when the VE decay reached 65%. The cell with a CO-CN@CF electrode exhibited an initial VE of 86%, and maintained stability (about 75% after 1000 cycles) during the entire cyclic process, whereas the cell with the pristine CF electrode declined from 75% to 65% at only 600 cycles, as shown in Figure 6a. Similarly, the EE slightly dropped from 84% to 73% in the cell with the CO-CN@CF electrode after 1000 cycles (over 400 h), whereas the cell with a pristine CF electrode decreased from 74% to 64% in just 600 cycles (about 150 h).

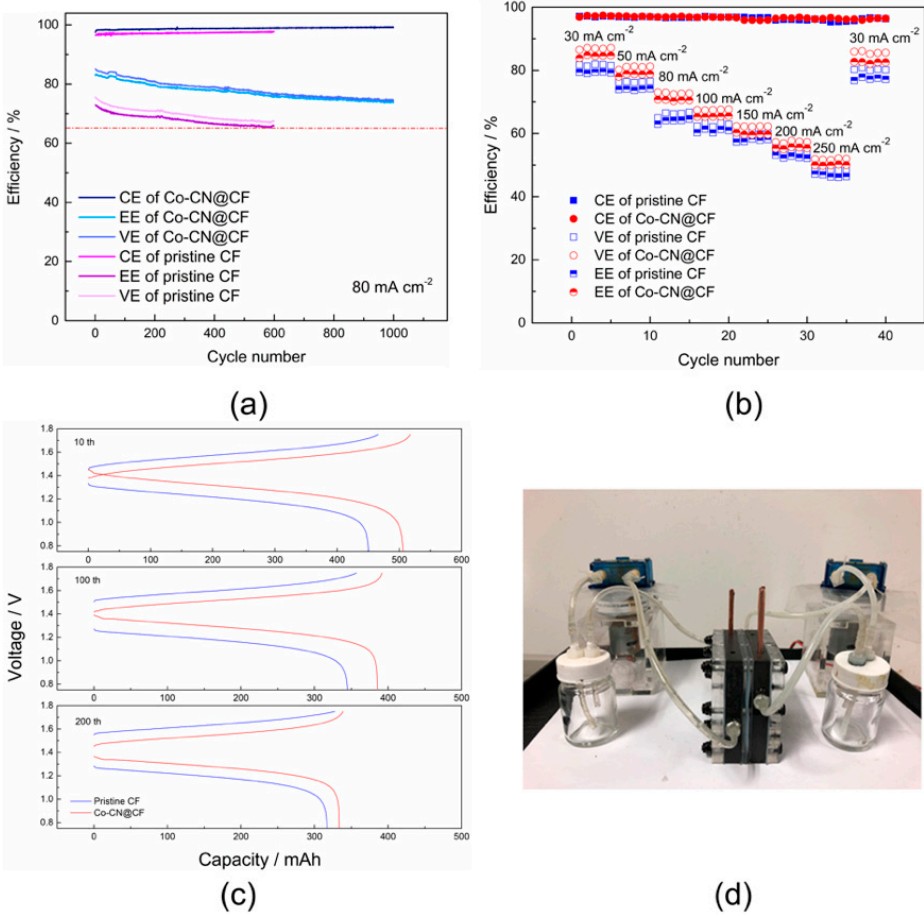

**Figure 6.** (**a**) Long-term stability, coulombic efficiency (CE), energy efficiency (EE), and voltage efficiency (VE) of the VRFB single cells with pristine CF and CO-CN@CF electrodes; (**b**) the rate performance of the VRFB single cells with pristine CF and CO-CN@CF electrodes at a current density of 30 to 250 mA cm$^{-2}$; (**c**) charge–discharge curves at 80 mA cm$^{-2}$ for the 10th cycle, 100th cycle, and 200th cycle; (**d**) the VRFB single cells.

Figure 5b presents the coulombic efficiency (CE), voltage efficiency (VE), and energy efficiency (EE) of the pristine CF and CO-CN@CF electrodes at a current density ranging from 30 to 250 mA cm$^{-2}$. The CE of the cell with the CO-CN@CF electrode showed high stability over the entire testing range, while that of the pristine CF electrode exhibited some fluctuation, especially at low current densities (e.g., 50 mA cm$^{-2}$). When returning from 250 to 30 mA cm$^{-2}$, the CE for the former recovered to the initial value, whereas the CE for the latter declined to a lower value. This implied that the Co-CN catalyst on the CF could keep the reaction stable in a large-scale current density, and that the wettability was also a key point for the VRFB reaction. Moreover, the efficiency of the cell with a CO-CN@CF electrode was higher than that of the pristine CF electrode throughout the rate performance test. The EE of the cell with a CO-CN@CF electrode still retained ≈52.6% at 250 mA cm$^{-2}$, as

compared to 48.7% for the CF electrode counterpart. This indicated that the Co-CN catalyst could improve the electrochemical performance. Figure 6c compares the charge–discharge curves of the cells with pristine CF and COvCN@CF electrodes at 80 mA cm$^{-2}$. Both cells exhibited almost the same charging profile. The cell with a CO-CN@CF electrode presented a higher discharge voltage and capacity.

## 4. Conclusions

ZIF-67 (200–300 nm) and CO-CN (100–200 nm) were prepared in situ with the CF; the surface area of the ZIF-67@CF increased, but the electric conductivity decreased due to the specialty of the 2-MI. After annealing at 600 °C for 2 h at a rate of 2 °C min$^{-1}$ under an Ar atmosphere, the ZIF-67@CF changed into Co-CN@CF. The ZIF-67 was the precursor of the Co-CN, as they kept the same stereochemical structure, but the Co-CN coalesced with CF, and the Co metal appeared after annealing in an inert atmosphere. With a unique structure and high efficiency catalyzed on the carbon felt, the Co-CN@CF exhibited excellent electrochemical activity toward the $VO^{2+}/VO_2^+$ redox couple in the VRFB, with an average cell voltage efficiency (VE) of 86% and an energy efficiency (EE) of 82% at 80 mA cm$^{-2}$, which was increased by more than 10% compared with the traditional carbon felt. VRFBs with a Co-CN@CF electrode also showed much better long-term stabilities (over 1000 cycles) compared with the batteries with pristine CF electrodes, because the redox-mediated catalysis of the Co-CN catalyst strongly bonded to the surface of the CF, and the $-OH$ on the surface increased the wettability of the CF.

**Author Contributions:** Conceptualization, L.Q. and J.S.; methodology, J.S.; software, J.S.; validation, J.S. and Z.L.; formal analysis, J.S.; investigation, L.H.; resources, L.H.; data curation, L.Q.; writing—original draft preparation, J.S.; writing—review and editing, J.S.; visualization, L.H. All authors have read and agreed to the published version of the manuscript.

**Funding:** This research received no external funding.

**Institutional Review Board Statement:** Not applicable.

**Informed Consent Statement:** Not applicable.

**Data Availability Statement:** The data cannot be provided.

**Acknowledgments:** The authors are grateful for the equipment support from Huang Jiamu's laboratory, School of Materials Science and Engineering, Chongqing University.

**Conflicts of Interest:** The authors declare no conflict of interest.

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
