# Peer review of "Preparation and Electrocatalytic Activity of a Cobalt Mixed Nitrogen 3D Carbon Nanostructure @ Carbon Felt toward an All-Vanadium Redox Flow Battery"

_applsci, doi:10.3390/app12052304_

Round 1

Reviewer 1 Report

A valuable work, with a clear presentation of the field and the scientific niche it addresses. Methods used are appropriate and relevant; they lead to clear conclusions.

A Question:

How is the cobalt content fixed on the carbon pellet controlled? As the recipe is presented, is it reproducible? 

Author Response

Response to Reviewer 1 Comments

Point 1: How is the cobalt content fixed on the carbon pellet controlled? As the recipe is presented, is it reproducible?

Response 1: We thank the reviewer for the constructive and helpful comments in improving the quality of our manuscript. The cobalt content fixed on the carbon flet could be controlled by the MOF (ZIF-67) content fixed on the carbon flet. And the MOF (ZIF-67) content fixed on the carbon flet could be controlled by the weight of carbon flet added into the reaction liquid. And those results are reproducible as the recipe. Thanks again.

Reviewer 2 Report

This is a very nicely presented paper which contains interesting results of high scientific quality. The material examined is well characterized and the electrochemical analysis is well developed. Scientific results are clearly presented and all relevant detail is available in the paper. References to the literature are both pertinent and are chosen well.

I recommend publication..

Author Response

Response to Reviewer 2 Comments

Response: We thank the reviewer for the constructive, kindness and helpful comments in improving the quality of our manuscript.

Reviewer 3 Report

The manuscript reported the carbon felt electrodes (Co-CN@CF) based on Co-MOF (ZIF-67) as the precursor and cobalt mixed nitrogen 3D carbon nanostructure for the application of VRFBs. With the unique structure and high-efficiency catalyst, the Co-CN@CF exhibits excellent electrochemical activity towards VO2+/VO2+ redox couples and energy efficiency (EE) of 82% at 80 mA cm-2, increased more than 10% contrast with the traditional carbon felt. VRFBs with Co-CN@CF electrodes also show better long-term stabilities over 1000 cycles.

I consider the content of this manuscript will definitely meet the reading interests of the readers of the Applied Sciences journal. However, the discussion and explanation should be further improved. Therefore, I suggest giving a major revision and the authors need to clarify some issues or supply some more experimental data to enrich the content. This could be a comprehensive and meaningful work after revision.

More detailed comments can be found in the PDF document.

Round 2

Reviewer 3 Report

I have carefully read the author's reply to the reviewer. I consider the author fully respected all the previous suggestions and comments proposed by me, and have made serious point-to-point replies and considerable revisions within the scope of the full text. It can be seen that the author worked very hard in the revision process.

I can no longer put forward any other suggestions and comments to improve this manuscript. The current version of the manuscript is acceptable to me.